# Weed Detection from Unmanned Aerial Vehicle Imagery Using Deep Learning—A Comparison between High-End and Low-Cost Multispectral Sensors

**DOI:** 10.3390/s24051544

**Published:** 2024-02-28

**Authors:** Anna Teresa Seiche, Lucas Wittstruck, Thomas Jarmer

**Affiliations:** Institute of Computer Science, Osnabrück University, 49090 Osnabrück, Germany; aseiche@uni-osnabrueck.de (A.T.S.); lwittstruck@uni-osnabrueck.de (L.W.)

**Keywords:** weed detection, self-built, drone, precision farming, Raspberry Pi, U-Net

## Abstract

In order to meet the increasing demand for crops under challenging climate conditions, efficient and sustainable cultivation strategies are becoming essential in agriculture. Targeted herbicide use reduces environmental pollution and effectively controls weeds as a major cause of yield reduction. The key requirement is a reliable weed detection system that is accessible to a wide range of end users. This research paper introduces a self-built, low-cost, multispectral camera system and evaluates it against the high-end MicaSense Altum system. Pixel-based weed and crop classification was performed on UAV datasets collected with both sensors in maize using a U-Net. The training and testing data were generated via an index-based thresholding approach followed by annotation. As a result, the F1-score for the weed class reached 82% on the Altum system and 76% on the low-cost system, with recall values of 75% and 68%, respectively. Misclassifications occurred on the low-cost system images for small weeds and overlaps, with minor oversegmentation. However, with a precision of 90%, the results show great potential for application in automated weed control. The proposed system thereby enables sustainable precision farming for the general public. In future research, its spectral properties, as well as its use on different crops with real-time on-board processing, should be further investigated.

## 1. Introduction

### 1.1. Motivation

With the rapidly increasing population around the world, diminishing cultivable areas, and unreliable farming conditions due to climate change, the pressure on agriculture is rising. In order to meet the growing demand for food in the future, innovative solutions that increase the efficiency of agriculture and are accessible to everybody are needed [1]. Weeds are the primary cause of yield reduction [2], as they compete with crops for nutrition, water, sunlight, and space [3,4]. The use of herbicides is the most common and efficient tool to control weeds but leads to irreversible ecological damage such as groundwater pollution, soil contamination, and biodiversity loss [5,6,7]. Given the spatial variability of weed cover, site-specific weed control can significantly reduce herbicide use by up to 39.2% [5] and is therefore essential for sustainable crop management. Unmanned aerial vehicles (UAVs) have recently proven to be an effective platform for weed monitoring, capable of flying close to crops and covering large areas in a short time [3]. A crucial first step for the application of targeted weed control measures, such as spot spraying, is the precise and reliable detection and discrimination of weeds and crops [8], as wrong weed detection information may lead to failure in weed removal or even cause crop damage [9].

### 1.2. Related Research

Recent studies have explored different methods for weed detection [10,11], ranging from computer vision image processing [9,12] and machine learning (ML) techniques [13,14,15,16,17] to state-of-the-art deep learning (DL) algorithms [18,19,20]. Since weeds and crops often share similar colors, textures, and shapes, the effective extraction of distinguishing features between crops and weeds is critical. Concerning this challenge, several studies have revealed the superiority of DL methods over traditional ML techniques [21,22]. As demonstrated by Ong et al. [23] in a case study on Chinese cabbage, a convolutional neural network (CNN) accurately detected weed by 92.41%, notably surpassing the performance of the opposing Random Forest model. Object detection and instance segmentation models have also recently outperformed the ML algorithm Support Vector Machine [24]. Further research has been conducted to evaluate the performance of various DL semantic segmentation networks such as a U-Net, SegNet, FCN, and DeepLabv3+, employing pixel-based image classification to differentiate between the background, crops, and weeds [3,25,26,27,28,29]. In a study using multispectral UAV imagery, a U-Net proved most efficient in terms of computational resources as well as model performance [27]. This is further demonstrated by the U-Net’s proficiency in real-time segmentation during on-board processing [30] and its successful application to a blurry UAV-based dataset of sorghum [28]. However, the precise detection of weeds relies not only on the choice of a high-performance algorithm but also largely on the applied sensor.

As Peña et al. [6] have shown, the use of multispectral cameras promises higher accuracy compared to RGB sensors, as they are able to capture additional plant characteristic spectral information and provide more robust imagery under varying lighting conditions [19,31]. Despite their advantages, commonly used high-end multispectral cameras for agricultural applications like the Parrot Sequoia+, Sentera Double 4K, Airinov MultiSpec 4C, and the MicaSense Altum, are usually very expensive (approx. EUR 15,000) and thus affordable to only a minority of users. To overcome these economic limitations, the Raspberry Pi Foundation’s customizable single-board computers, combined with the corresponding RGB and infrared camera modules, provide a cost-effective alternative to commercial multispectral camera systems [32]. These so-called Raspberry Pis are characterized by their light weight, compactness, and high-performance system on chip, functioning as a fully fledged computer [33]. Unlike the Arduino microcontroller board, they offer high computing power and memory, which makes them suitable for a broad spectrum of scientific and engineering applications [34]. In 2017, Pagnutti et al. [35] investigated the radiometric characterization of raw-data format imagery acquired with the Raspberry V2.1 camera module, laying the foundation for comparison with other sensors and further application-oriented research. However, there are only a few studies in which a Raspberry Pi-based multispectral camera has been developed and applied in agricultural contexts. Pioneering approaches in this field were conducted by Dworak et al. [36], Döring et al. [37], and Sangjan et al. [38], who implemented Raspberry Pi-based multispectral camera systems for NDVI (normalized difference vegetation index) calculation, while Belcore et al. [33] and Döring et al. [37] have demonstrated the successful integration of a similar system into a UAV for automated image acquisition. Regarding weed and crop identification, initial attempts using RGB sensors on robots or ground stations have been made [4,39]. However, to date, there are no studies in which a low-cost multispectral camera has been employed for specific challenges in precision agriculture.

### 1.3. Goals and Structure

Further investigations are fundamental in assessing the efficacy of a Raspberry Pi-based multispectral camera system in weed management. To address this research gap, this paper proposes an innovative Raspberry Pi-based low-cost camera system (LCS) developed for agricultural applications and capable of autonomous multispectral image capture on a UAV. Aiming to offer a reliable alternative to commercial sensor systems, the performance of the LCS in accurately detecting weed was benchmarked against the high-end multispectral camera MicaSense Altum. The study was conducted using UAV data in early-season maize, being a valuable study case for its popularity and vulnerability to weeds [6]. Different from the approach in previous work [40], this research does not focus on developing a novel classification algorithm, which is why the well-established U-Net architecture was employed for weed segmentation. The U-Net’s strong feature learning capabilities and automatic feature extraction allow for easy scaling to new datasets, making it highly suitable for real-world applications [10,22,41]. The pixel-based classifications generated by this dense semantic segmentation framework serve as a valuable data source for autonomous weed control based on precise localization information.

In summary, this research aims to

Present the hardware and software development of a multispectral LCS for weed detection in agriculture.Compare the performance of the proposed multispectral LCS against the trusted MicaSense Altum in a real-world scenario. The classification accuracy of both sensors will be evaluated in the context of a practical application using different evaluation metrics.Identify and discuss possible weaknesses of the LCS to determine suitable use cases and outline a path for further development of the system.

## 2. Materials and Methods

### 2.1. Data Acquisition

#### 2.1.1. Sensor Description

In this work, the well-performing high-end multispectral camera MicaSense Altum was compared with a self-built multispectral LCS. Thereby, the Altum served as a benchmark due to its successful application in numerous studies [42,43,44]. As a sensor specialized in agricultural analysis, the Altum integrates a radiometric thermal camera alongside five discrete narrow bands, enabling the simultaneous capture of advanced thermal, multispectral, and RGB imagery in a single flight [44]. The spectral bands encompass the channels blue at 475 nm, green at 560 nm, red at 668 nm, red edge at 717 nm, and NIR at 842 nm (see Table 1). Its high spatial and radiometric resolution makes it the ideal sensor for machine learning plant-level applications such as early-stage weed detection [45].

The proposed LCS employs the Raspberry Pi camera module V2 equipped with a Sony IMX219 image sensor for RGB imaging and a similar version sensitive to infrared wavelengths. Accordingly, it offers red, green, blue, and NIR channels as a multispectral system. However, unlike the Altum system, which captures each RGB channel using distinct sensors, the LCS records those through a single sensor. This is reflected in its spectral sensitivity, with bandwidths in the visible spectrum extending to approximately 100–150 nm, as opposed to those of the Altum system, at 14–32 nm [44]. Additionally, the central wavelengths in the LCS are shifted approximately 40–90 nm towards shorter wavelengths [35] (see Figure A1 and Table 1). Given its uncommon spectral characteristics, there is great research interest in the sensing capabilities of this pioneering system. Although the field of view and spatial resolutions of both camera systems differ due to their inherent hardware properties (see Table 1), they yield comparable GSDs, approximately 0.55 cm/px.

#### 2.1.2. System Hardware LCS

The proposed system is based on the single-board computer Raspberry Pi 4B [46]. A connected powerbank with a 20,000 mAh capacity guarantees a recording time of several hours. Since the Pi 4 only has one camera port, it was further linked to two computers, namely Pi Zeros, via the plug-on board ClusterHAT [47] (see Figure 1). The Zeros are particularly small and inexpensive and serve as connections for the camera modules. The installed visible and infrared sensors are identical in construction, with the difference that, on the so-called “NoIR”, no IR-cut filter was installed above CMOS. The subsequently added bandpass filter allows the chip to be sensitive only to the infrared spectrum. The sensor specifications can be taken from Table 1. To ensure a similar shooting angle for both Pi cameras, the V2.1 lenses were detached from their circuit boards and connected to them via an extension cable. The camera’s housing was modeled using Blender software and then 3D-printed. This ensures that the device is ventilated while being protected from environmental influences. 

#### 2.1.3. System Software LCS

The Raspberry Pi 4 as well as both Pi Zeros are equipped with SD cards on which Raspbian operating system images (available at https://dist1.8086.net/clusterctrl/buster/2020-08-20/, accessed on 18 February 2024) are installed. The Pi 4 acts as the Controller Pi and creates an IP-subnet for the Pi Zeros via the installed CNAT image, which allows access to the sub-Pis via Secure Shell Protocol (SSH). All captured images were stored on the Controller Pi. To ensure the exact same time of capture, a server application was implemented on each of the sub-Pis so that the Controller Pi serving as a client could simultaneously trigger the cameras and request the images. Images were taken with a delay of two seconds between each shot, same as with the Altum system. On each Pi Zero, a Python script determined the settings during the recording, which were adjusted automatically. Only the white balance of the NIR camera was set statically to the surrounding illumination using awb_gains = (0.5, 2.0).

#### 2.1.4. UAV Platform and Study Site

Aerial flights were conducted under cloudy conditions in a maize (*Zea mays* L.) field in Lower Saxony, Germany. The flight was carried out during the two-leaf stage of the maize, i.e., at a BBCH stage of 12. This describes the most sensitive phenology stage of the plant and is therefore a common time for initial weed control measures [48]. The flight altitude of 12 m resulted from a trade-off between flight time and spatial resolution that ensures visibility of the weeds in the early vegetation stage. Unlike the Altum camera, the sensors of the LCS do not feature autofocus, which is why the lenses were manually focused to the appropriate distance before flight. As the focus is highly sensitive to even slight adjustments, it was fine-tuned through an iterative trial-and-error process to attain the best possible sharpness. Both camera systems were attached to the UAV (DJI M2010) side by side (see Figure 2) in order to capture the photos under comparable conditions.

### 2.2. Data Analysis

#### 2.2.1. Preprocessing

To align the images of the individual channels, they were geometrically transformed using image correlation (Enhanced Correlation Coefficient Maximization) [49]. Subsequent cropping of the image edges by approx. 25% effectively removed any distortions, vignetting caused by the bandpass filter, and overlapping of the images (see Figure 3 Section I). Since the available DL architecture is primarily designed for a three-channel input, it has been limited here to the red, green, and NIR bands, being most relevant for vegetation detection [27,50,51,52] and commonly used in comparable studies [53,54,55,56]. The NIR channel is less sensitive to differences in solar incidence, contributing to a more robust classification with dynamic weather conditions [57]. The red, green, and NIR information were provided separately rather than combined into the NDVI for a straightforward classification input. Radiometric calibration was intentionally not performed to simplify preprocessing. Furthermore, the focus of the study lies in the comparison of the classification performance and not the spectral properties. A vegetation mask served as the basis for subsequent annotation of the images. This follows well-established practices and is similarly applied in the studies evaluated [31,58,59]. To create a vegetation mask based on which to label the training images similar to [31], the NDVI [60] was extracted from the layerstack. This index was used to distinguish between vegetation and soil. An intensity histogram clustering algorithm, Otsu’s method [61], was used for individual threshold selection. Figure 4 displays the corresponding intermediate steps of the preprocessing with the resulting binary mask. These were manually annotated by the author using GIMP image editing software. Thereby, the original image was placed under a mask for direct comparison and all pixels marked as vegetation were divided into the classes weed or crop, while the background pixels were classified as soil. No additional correction of the vegetation mask, such as removing oversegmentations or adding weeds, was performed. This kept inaccuracy due to human influence low and comparable across all training data. In addition, a practicable and easily comprehensible workflow was aimed for. With the intention of evaluating the LCS as an alternative to the Altum system, it was also important to annotate the training data of both systems independently.

A total of 75 images were labeled per sensor, with an extension of 1164 × 920 pixels. These images were then divided into 224 × 224 pixel tiles, yielding a total of 1500 training patches. As a further preprocessing step, the images were loaded into a range of [0, 1] and normalized according to the following: mean = [0.485, 0.456, 0.406], std = [0.229, 0.224, 0.225] for compatibility with the pre-trained segmentation model employed for training. In total, 20% of the data was retained for testing and 80% was used for the training process.

#### 2.2.2. Evaluation of Training and Testing Data

It must be noted that the accuracy of the vegetation mask-based annotation largely depends on the spectral and spatial resolution of the source image. This is because precise separation of red and NIR bands is fundamental for NDVI calculations, while low spatial resolution can cause mixed pixels of vegetation and soil, distorting the plant outlines [62,63]. In order to qualitatively assess the resulting inaccuracy in the ground truth, a subset of five vegetation masks of the same referenced field sections of both sensors were additionally corrected manually. Pixels that were incorrectly masked by the index were added accordingly. In case of doubt, the images of the Altum sensor served as a reference for the correction of both the Altum and LCS-based masks. The considered image sections were chosen heterogeneously in order to be representative of the whole field. A subsequent descriptive analysis of the sources of error allowed the accuracy of the mask to be estimated for each class in reference to the application background (see Figure 3 Section II).

#### 2.2.3. Semantic Segmentation

The annotated images were subsequently fed into the U-Net (see Figure 3 Section III). This state-of-the-art segmentation framework was originally introduced by Ronneberger et al. [64] for biomedical image segmentation but has also been successfully used for weed classification tasks in recent years [3,26,27,28,65]. As an encoder–decoder network, the architecture is based on that of an FCN, where a CNN is first placed upstream as a backbone (feature extractor) (see Figure 5). In this study, the well-established ResNet-50 [66], consisting of convolutional and max pooling layers, was used for this purpose. In order to maintain the 2D structure of the images, the last fully connected layers in the CNN architecture were replaced by fully convolutional layers, which were upsampled step by step in the corresponding decoder stage. Unlike FCN, learnable filters were used instead of fixed bilinear interpolation. In addition, skip connections transferred information from the encoder layers to the corresponding decoder layer. This returned precise localization using transposed convolutions [27]. Existing research proves that this model achieves good segmentation results even with little training data by relying on a vigorous data augmentation pipeline [28]. The first concatenation level allows for an arbitrary number of input channels. Using a softmax function, a probability distribution was created over the possible classes for each pixel in the last layer. The class was then assigned according to the highest probability.

The implementation was carried out using the PyTorch framework. The feature extractor was initialized with weights pre-trained on ImageNet [67] to take advantage of transfer learning [66]. The batch size was limited to 4, being the maximum achievable with the given patch size due to hardware constraints. Training spanned 10 epochs, applying a learning rate of 0.001 for the initial 5 epochs and reducing to 0.0001 for the latter half to prevent overfitting [68]. To improve the convergence of the optimization process, the Adam optimizer [69] was employed. 

#### 2.2.4. Evaluation Metrics

The predictions of the U-Net on the retained test dataset were compared pixel by pixel with the corresponding annotated test data, which served as a ground truth (see Figure 3 Section III). For each image, the Jaccard index, precision, recall, and F1-score were calculated. Finally, the results were averaged over the entire test dataset. The metrics applied correspond to those that are most common in this field of research.

The Jaccard index (also known as the Jaccard coefficient or InterSection over Union) is an established metric used to evaluate the performance of a segmentation algorithm [58]. It is defined as the size of the interSection of the predicted and ground truth segmentations divided by the size of the union of the predicted and ground truth segmentations. The utilized Jaccard score computes the average of the Jaccard index, between pairs of label sets. The average is weighted, considering the proportion for each label in the dataset, to account for imbalances.The calculations of recall, precision, and F1-score were performed for each class separately. Recall is a measure of the ability of the model to correctly identify positive instances in a dataset, whereas precision measures the proportion of instances identified as positive by the model which are indeed positive. The F1-score combines these two metrics as a harmonic mean in a single score [70].To better comprehend the misclassifications, a confusion matrix (CM) was computed. The matrix contains the normalized number of true positive, true negative, false positive, and false negative predictions made by the U-Net for each class.

## 3. Results

### 3.1. Qualitative Results of the Training and Testing Data

Figure 6 and Figure A2 in the appendix illustrate the posterior correction of the annotated vegetation masks, using the corresponding Altum source images as a reference. These manually created ground truths were utilized to qualitatively evaluate the accuracy of the training and test datasets.

Both the Altum and LCS-based vegetation masks captured all maize plants and mapped their structure. However, the LCS data resulted in slightly oversegmented and less precise boundaries. Gaps within vegetation were accordingly effectively detected in the mask on the Altum data, whereas these were incorrectly included in the vegetation class based on the LCS data. Additionally, finer leaf ends of maize were only accurately masked in the Altum data, with these details often being misclassified as soil in the LCS masks. Regarding the misclassifications marked in red, it becomes visible that the Altum masks also occasionally fail to classify the leaf ends of maize as vegetation initially. When correcting the LCS-based masks, broader outlines were added to include coarser leaf ends. Narrow leaf sections, not depicted on the Altum vegetation mask, were partially captured in the LCS-based mask due to broader outlining.

In the Altum images, the weed class was mapped accurately without any exceptions. Even weeds consisting of less than 10 pixels were recorded as vegetation. These weeds were no longer clearly recognizable as vegetation to the human eye on the LCS images and were accordingly classified as soil via the index. As indicated in the figure, weeds smaller than 30 pixels were generally not identified as weeds in the annotated LCS mask. Moreover, larger weeds overlapping with maize were partially and erroneously assigned to the maize class in the annotations. Overall, while both sensors exhibit varying levels of detail in depicting plant structure, they accurately represent the majority of individual plants as vegetation, leading to precise annotations.

### 3.2. Quantitative Evaluation of the U-Net Predictions

The overall detection accuracy of the U-Net, measured by the Jaccard score, amounts to 96.75% on the Altum dataset and 93.27% on the LCS dataset. Figure 7 illustrates the prediction accuracies of the U-Net for the classes soil, crop, and weed. The accuracy measures recall, precision, and F1-score consistently indicate a 99% accuracy for the soil class based on the Altum data. The predictions with the LCS source data are insignificantly lower, ranging between 98 and 99% accuracy. The crop class shows greater sensor-related differences. With 87% vs. 97%, the LCS performed 10 percent points (pp) lower in precision. However, recall accuracy remained high for both sensors, at 95% and 96%, respectively. The weed class shows a significant variance across different evaluation metrics. The recall for the U-Net using LCS data recorded the lowest accuracy at 68%, which is 7 pp lower than that of the Altum data. Precision is considerably higher, at 92% for Altum images and 90% for LCS images. As expected, the F1-score represents the harmonic mean of these results.

The CMs depicted in Figure 8 provide information about the pixel-based class assignment of the neural network (NN). For both the Altum and LCS cameras, the misclassifications in soil are negligible. The maize class was likewise accurately classified on both datasets with high probability (~95%), while the predominant source of misclassification for this class is the confusion with soil. The weed class exhibits more pronounced detection inaccuracies based on both camera systems. Regarding the Altum images, 12.16% of the pixels have been incorrectly assigned to crop and 12.95% to soil, respectively. For the LCS images, 14.90% of weed pixels have been assigned to crop and 17.46% to soil.

### 3.3. Qualitative Evaluation of the U-Net Predictions

Figure 9 and Figure A3 in the appendix display the predictions of the U-Net on the hold-out test dataset. These are in opposition to the corresponding initial UAV images of the compared sensors and the annotated input provided to the NN. The quality of the LCS images differs not only in the displayed coloration but also in the spectral resolution and noise and blur ratio in comparison with the Altum images. The resulting inequality of the annotated training and test data have already been evaluated in Section 3.1. Regarding the Altum-based dataset, the output of the U-Net (blue background) aligns closely with the annotated vegetation mask (black background). In particular, the structure of the maize plants has been detected correctly on a pixel level. The weed shapes in the predictions also mostly mirror the ground truth, with occasional gaps in the plants and some weeds, especially those overlapping with maize, misidentified as crops. Other misclassifications occurred with a minority of the youngest weeds, which were not detected by the U-Net and thus assigned to the soil class. For the LCS source images, the predictions reflect the slight imperfections in the annotated input, resulting in less detailed representations of plant structures compared to the Altum-based predictions. Nonetheless, all maize plants and the vast majority of the weeds were also correctly classified. Notable misclassifications in the examined subset primarily occurred where weeds overlapped with maize plants, often leading to partial correct weed classification and partial misidentification as maize. Similar to the Altum dataset, there was further confusion in identifying smaller weeds, which is more pronounced in the LCS images.

## 4. Discussion

The accurate segmentation of crops and weeds with a cost-efficient sensor system and straightforward workflow is currently of great interest in precision agriculture. In this scientific work, a respective system was introduced and evaluated against a trusted high-end system with the intention of using it in weed management. The NDVI-derived vegetation mask for the generation of training data differs between the sensor-dependent datasets in pixel precision. Particularly maize has been slightly oversegmented on the LCS images, which can be explained by the ambiguous pixels, especially at the plant edges. Since the GSD of both systems is the same up to 0.55 px/cm, the reasons for the poorer image quality of the LCS are likely to lie in other factors. The slight blurring especially with the NIR camera is possibly due to a less precise manual focus. The absence of a display and the fine nature of the hardware complicated the manual focus setting of the lenses, making an autofocus a significantly more convenient option for the application. The higher noise ratio (see Figure 4) is presumably caused by lower-quality lenses that do not offer the same sharpness, contrast, as well as color accuracy as those of the Altum system. Additionally, the spectral responses of the Altum system are slimmer and more clearly separated between the examined bands (see Figure A1), resulting in sharper delineating index values between vegetation and soil. Accordingly, weeds consisting of a few pixels were captured by the index. Those smallest weeds, however, do not stand out clearly on the LCS images and were thus misclassified as soil via the threshold. Especially in the NIR channel, the gray-level difference between vegetation and soil of the low-cost sensor is less pronounced, which complicated the discrimination of classes for small segments based on the NDVI. However, since these misclassifications primarily occurred only for the smallest weeds on the LCS imagery, the annotations still serve as a reliable ground truth.

The predictions of the U-Net demonstrate high overall accuracies on both datasets, 93.3% (LCS) and 96.8% (Altum), respectively, which clearly exceed the performance of a study with a comparable dataset, 88% [29]. However, class-specific differences in accuracy are observed, with weeds showing the most misclassification. This is consistent with the results of other research [26,27,31] and can be partly explained by the strong class imbalance. Because there are a lot more crop- and soil-labeled pixels than weed-labeled pixels, the model might be more sensitive to detecting crops. Along with this, the values of the confusion matrix show a tendency to underestimate the weed class (see Figure 8). This occurred based on both sensor systems, whereas the recall with respect to weed is for the LCS, with 68%, which is a bit lower, presumably due to the spectral resolution. Nevertheless, the high precision values (90%) provide the user with reliable weed detection using the LCS. However, to account for class imbalance in the following work, an adjusted loss function (e.g., focal loss) should be applied.

The qualitative evaluations on the predictions of the U-Net reflect the class-specific strengths and weaknesses of the sensors. The segmentation detail corresponds to that of the training data, which is why minor misclassification occurred at the plant edges, based on the low-cost images. Such inaccuracies may be mitigated through the previously discussed methods aimed at refining the preliminary vegetation masks, especially by improving focus on the LCS. However, considering the context of real-world site-specific weed management, this minor oversegmentation is deemed irrelevant, as the general plant shape and its precise location are sufficient to generate accurate weed maps [28]. The overlap between weed and crop caused further confusion, which explains the tendency to overestimate the maize class. This is a common challenge in weed detection [10,58] and can possibly be optimized with the adjustment of the segmentation model. Apart from this, the outputs of the model differ with respect to very small weeds, which were not reliably detected on the LCS. This is known as a general bottleneck in DL approaches [28] and is likely related to a number of influences that cause poorer image quality of the LCS, determined by lens quality, spectral resolution, camera focus, etc. Further research should therefore aim to explore the improvement potential of the LCS by, for instance, evaluating alternative lenses or automatic focus. Additionally, investigations regarding the filter on the NoIR sensor should be undertaken to optimize the contrast in the NIR channel [36]. In this respect, it must be considered that the wavelength ranges of the two camera systems differ in the NIR range, which influences the detection accuracy due to changing reflection properties of the vegetation. Customizing the bandpass filter by aiming at a wavelength range similar to that of the Altum camera could accordingly improve the information of the LCS NIR channel. In addition, detailed research is needed on the reflectance properties of the Raspberry Pi NoIR sensor, including a radiometric and spectral calibration.

Overall, the performance of the LCS is only slightly weaker than that of the high-end model in terms of weed detection (F1-score of 76% vs. 82%), showing great potential for a wide range of applications. The accurate prediction of weed patches makes the system suitable for spot spraying, for which in the context of further work, the extraction of the image localization via differential GPS should be implemented. Furthermore, flying with a larger overlap would allow the images to be aligned into an orthophoto, providing valuable data in regard to weed mapping. The predictions are also suitable as a basis for mechanical hoeing due to the high recall, although the precision decreases with the LCS on small weeds. Especially in application scenarios with a significant distance between the sensor and the object of interest, similar to the experimental setup examined in this study, the LCS may underperform compared to the Altum system. Besides filter optimization for better spectral resolution, weed detection accuracy is therefore likely to increase further with lower UAV altitude and correspondingly higher spatial resolution. Integrating the presented system into a weeding robot, as an advancement of the methodology proposed by Chechlinski et al. [4], would thus be of great research interest. To address the limitations of the LCS in detecting small weeds, alongside the suggested enhancements to both setup and hardware, it is essential to consider the necessity of regular hoeing. This allows for the identification of weeds that were overlooked previously. Given this context, the disparity in precision relative to the high-end system has a negligible effect in practical applications.

Moreover, the Raspberry Pi serves as a comprehensive computing device which, unlike the Altum system, enables on-board processing of image data. Through modifications to the U-Net architecture, as demonstrated by Chechlinski et al. [4] and Zou et al. [71], it thereby becomes possible to achieve local, real-time weed detection. With this vision for future research, it is crucial to underscore the LCS’s potential for UAV applications. Integrating the proposed sensor with a suitable drone would provide a standalone low-cost system and thus address a larger number of users globally than cost-intensive weeding machines. In this context, the LCS needs to be tested in different crops and study areas to evaluate its generalization capabilities.

## 5. Conclusions

In this paper, the usability of a multispectral low-cost UAV-based sensor system in the practice of weed detection was demonstrated. The implemented DL model achieved satisfying overall accuracies, with 93.27% based on the proposed LCS imagery and 96.75% on the high-end sensor, which was used as a benchmark. The recall of the weed class on the LCS dataset declined to 68%. However, the simultaneously high precision of 90% ensures a reliable localization of the weeds from the user’s perspective. Further, a qualitative analysis indicates that the general form of most crops and weeds can be detected accurately, with minor misclassification occurring with small weeds and overlaps. To address this challenge, future research should focus on the radiometric and spectral analysis of the Raspberry Pi sensors to further improve the spectral resolution of the system using appropriate filters and calibration. In addition, the variation in the lens and the implementation of an automatic focus on the LCS might offer room for improvement. Overall, the LCS-based prediction is a very good representation of the ground truth weed distribution and shows great potential for applications such as spot spraying and mechanical weeding. The implementation of real-time on-board processing is now essential to achieve sustainable, targeted, and affordable weed control with minimum human intervention.

## Figures and Tables

**Figure 1 sensors-24-01544-f001:**
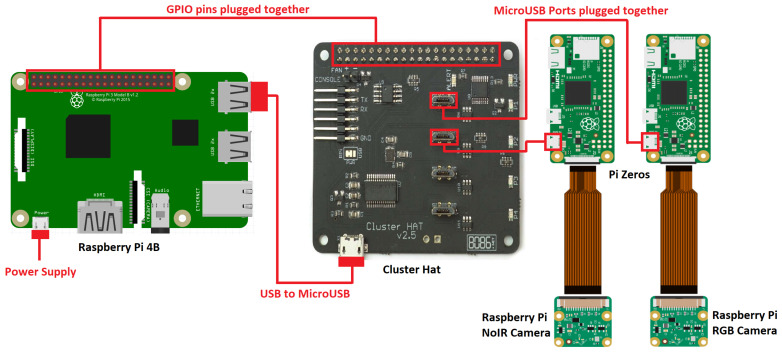
Wiring diagram of the proposed LCS.

**Figure 2 sensors-24-01544-f002:**
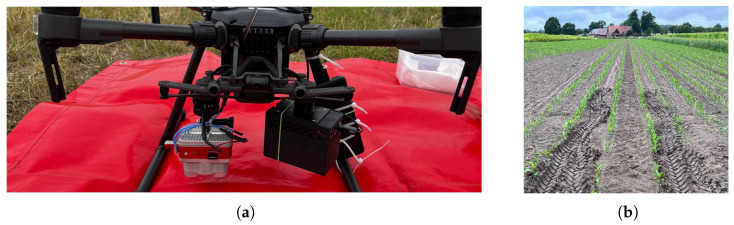
Flight setup and location. (**a**) Sensor mounted (**left**: Altum, **right**: LCS). (**b**) Investigated agricultural field.

**Figure 3 sensors-24-01544-f003:**
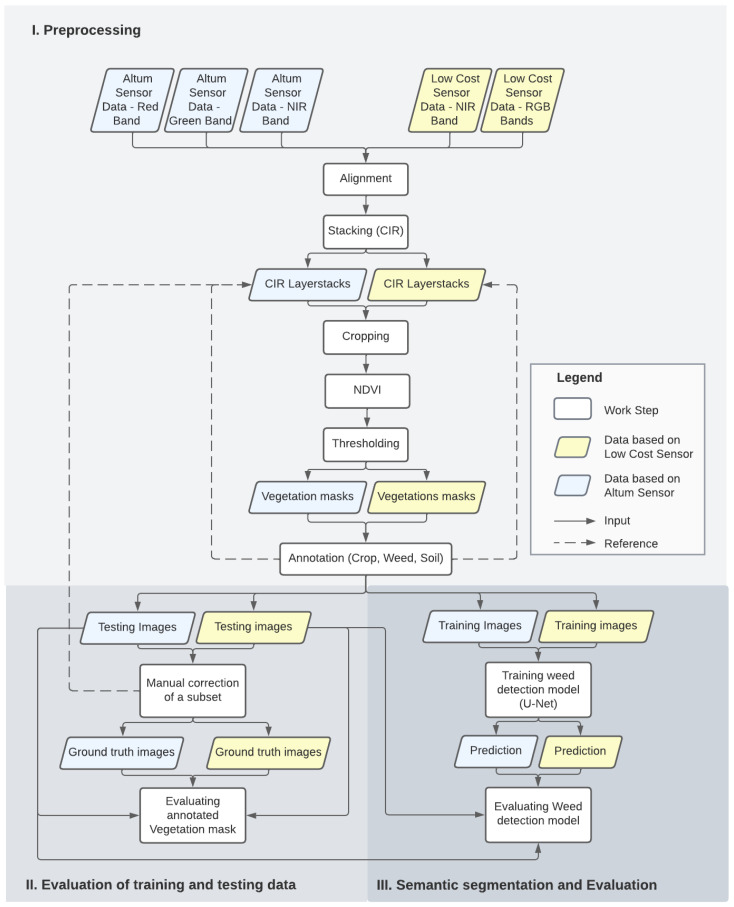
Flow chart of performed work steps, subdivided according to paper structure.

**Figure 4 sensors-24-01544-f004:**
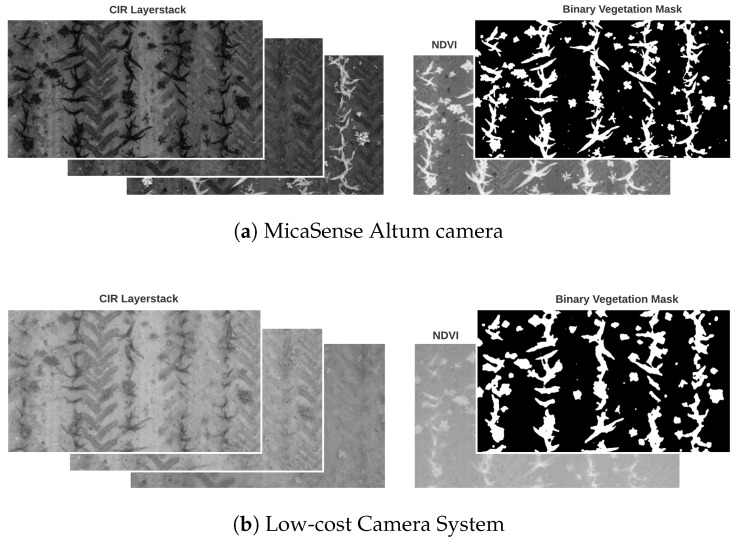
Intermediate results from input data via NDVI to vegetation mask.

**Figure 5 sensors-24-01544-f005:**
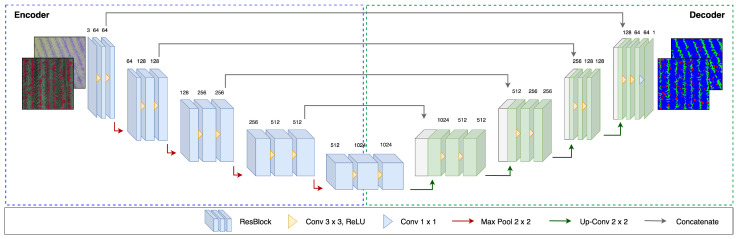
The U-Net architecture with skip connections, where ResNet50 is used as an encoder and transposed convolutions are used in the decoder for upsampling the result after bottlenecking. The encoder outputs a 7 × 7 grid size, with the final convolutional layer’s output matching the input dimension.

**Figure 6 sensors-24-01544-f006:**
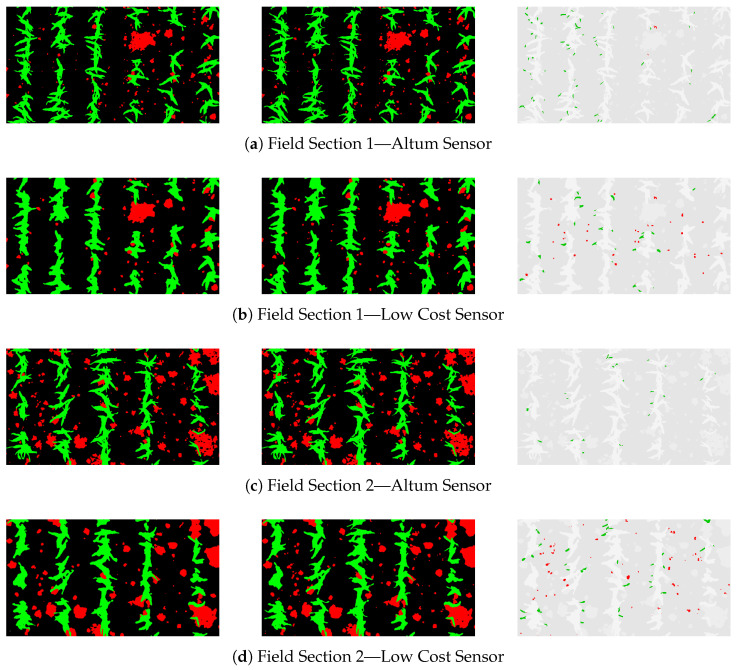
Qualitative evaluation of training and testing data inaccuracy. **Left**, the annotated vegetation mask. **Middle**, the manually corrected annotated vegetation mask. **Right**, the corrected pixels highlighted, with red indicating weed and green denoting maize.

**Figure 7 sensors-24-01544-f007:**
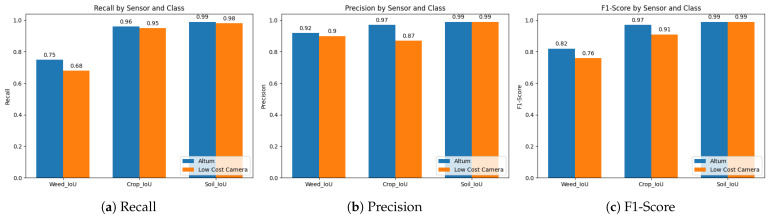
Quantitative evaluation results of the U-Net predictions on the hold-out test data.

**Figure 8 sensors-24-01544-f008:**
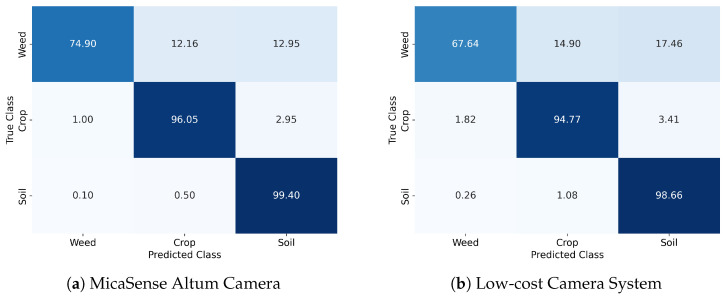
Normalized confusion with pixel-based classification results on the hold-out test set in percent. (**a**) Showing the results on the Altum system-derived dataset and (**b**) showing the low-cost system results. The darker the blue shade, the higher the proportion of pixels assigned to that class.

**Figure 9 sensors-24-01544-f009:**
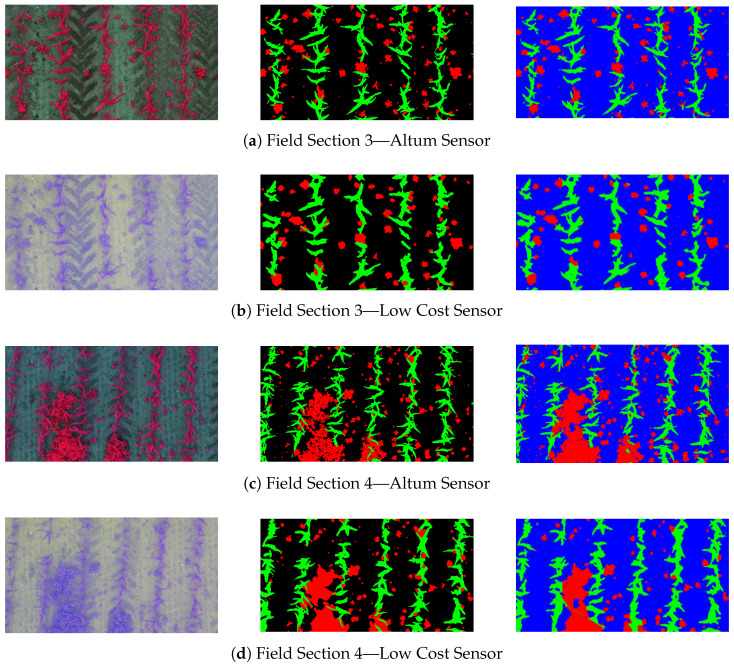
Sample qualitative results achieved on the two investigated sensor systems. **Left**, the corresponding UAV images. **Center**, the annotated vegetation mask as hold-out testing images. **Right**, the predictions of the U-Net.

**Table 1 sensors-24-01544-t001:** Overview of investigated sensors with lens and imagery information.

	MicaSense Altum	Raspberry Pi 4 V2 Sony IMX219 RGB + NoIR
Resolution	2064 × 1544 px	2592 × 1944 px
Field of view	48° × 36.8°	53.50° × 62.2°
GSD (at 12 m altitude)	∼0.53 px/cm	∼0.55 px/cm
Center Wavelength	Blue: 465 nm, Green: 560 nm, Red: 668 nm, NIR: 842 nm	Blue: 450 nm, Green: 520 nm, Red: 600 nm, NIR: 750 nm
Price	∼15,995.00 €	∼500.00 €

## Data Availability

The raw data supporting the conclusions of this article will be made available by the authors on request.

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
