# Peer review of "Weed Detection from Unmanned Aerial Vehicle Imagery Using Deep Learning—A Comparison between High-End and Low-Cost Multispectral Sensors"

_sensors, 2024, doi:10.3390/s24051544_

Round 1

Reviewer 1 Report

Comments and Suggestions for Authors

This manuscript offers a commendable analysis, drawing a valuable comparison between a widely used off-the-shelf multi-spectral camera and a bespoke, budget-friendly multi-spectral alternative. The potential impact on simplifying technology adoption in agriculture is particularly noteworthy. While the manuscript successfully delivers insightful comparisons and engaging discussions, a meticulous revision is warranted. This should address grammatical errors, such as inappropriate article usage and verb form discrepancies, ensure consistency in significant numbers, incorporate the appropriate use of acronyms, and rectify any instances of non-English characters that may have slipped through.

 Specific comments are below:

L83: It's advisable to replace "an" with "a" when referencing UAV to align with pronunciation rather than the presence of the vowel.

L128: You can abbreviate GSD since it has already been defined in the abbreviations section, eliminating the need for the full words.

Table 1: Please replace "Prize" with "Price" and use the abbreviation GSD for brevity.

Figure 1: I would advise against using "Schematic diagram," as it closely resembles "Pictorial diagram." Opting for a more distinct term will contribute to clarity and precision in communication.

L144: Please clarify what all three Raspberry Pis are, such as a Pi 4 and two Pi Zeros.

L155: When employing binomial nomenclature, refrain from italicizing the standard abbreviation "L." for "Linnaeus." This practice adheres to the established conventions of biological classification and ensures accurate representation in scientific documentation.

Figure 2: Consider refining the title to encapsulate more meaningful information, providing a clearer and concise representation of the content within the document.

Figure 3: This appears to lean more towards a flow chart, with the parallelograms likely representing data or input/output points.

L302: Certainly, providing clear directional cues enhances readability. Please specify which camera you are currently explaining, do not assume all the readers will easily follow your flow without any clear directions. In this case, I believe that commencing words, such as “For both Altum and LCS cameras,”, should be placed prior to the result.

L304: Please clarify which of ¾ and ¼ are recognized as what. If my assumption in L302 is right and you rectified, you don’t need to do this. I guess ¾ of misclassification of the crop (maize) was classified as soil, do not use background as soil is the right category in the manuscript, and ¼ of misclassification of the crop was classified as weed from both cameras which bring a lot of confusion as both cameras were not mentioned, please rectify accordingly.

L307-308: Use exactly same data compared to Figure 8.

L355: Double check the matching word of exceed, I understand “accuracies”. If it is, it should be “exceed”.

Figure A1: Do not use any non-English words in the legend, such as Blau, Graun, and Rot.

-END-

Comments on the Quality of English Language

This was already mentioned in my critics.

Reviewer 2 Report

Comments and Suggestions for Authors

This paper addresses the significant challenge of efficient and sustainable cultivation in agriculture, particularly under the constraints of changing climate conditions. The focus on developing a low-cost, multispectral camera system for weed detection and its comparison with a high-end alternative is both timely and relevant.

Summary of the Paper:

The paper presents a novel, self-built, low-cost multispectral camera system designed for weed and crop classification using UAV data sets. The research evaluates this system against the high-end MicaSense Altum using a U-Net for pixel-based classification. The approach includes an index-based thresholding method followed by annotation to generate training and testing data. The results show a promising F1-score and recall for weed classification, with the low-cost system slightly underperforming compared to the high-end system. Misclassifications were noted, particularly in small weeds and overlaps. However, the precision of the low-cost system is commendable. The paper concludes by highlighting the potential of the system for automated weed control and sustainable precision farming.

Strengths of the Paper:

Relevance and Originality: The paper addresses a crucial need in precision agriculture and offers an innovative solution that is accessible to a broader user base.

Methodological Rigor: The experimental setup, including the use of two different sensors and a systematic approach for data collection and analysis, is well-designed.

Potential Impact: The development of a low-cost alternative to high-end systems could significantly impact sustainable farming practices.

Areas for Improvement:

Comparative Analysis: While the comparison between the two systems is insightful, a more detailed discussion on the specific scenarios where the low-cost system may underperform would be beneficial.

Limitations: The paper briefly mentions misclassifications and oversegmentation but does not delve into how these could be mitigated or their potential impact on practical applications.

Future Work: The suggestions for future research are appropriate; however, elaborating on how the proposed system could be adapted or improved for these future applications would add value.

Conclusion:

In conclusion, the manuscript is well-written, addresses a significant challenge in precision agriculture, and presents a novel solution with substantial potential benefits. The study is methodologically sound, and the results are promising, albeit with some limitations that should be addressed in future work.

I recommend the acceptance of this paper for publication, provided that the authors consider the points raised for improvement in a revised version.

Comments on the Quality of English Language

ok
